# Male Victims of Sexual Abuse: Impact and Resilience Processes, a Qualitative Study

**DOI:** 10.3390/healthcare11131868

**Published:** 2023-06-27

**Authors:** Léa Poirson, Marion Robin, Gérard Shadili, Josianne Lamothe, Emmanuelle Corruble, Florence Gressier, Aziz Essadek

**Affiliations:** 1Interpsy Laboratory, University of Lorraine, 54015 Nancy, France; 2Department of Adolescent and Young Adult Psychiatry, Institut Mutualiste Montsouris, 75014 Paris, France; 3CESP, INSERM U1178, Team PsyDev, University Paris-Saclay, UVSQ, 94275 Gif-sur-Yvette, France; 4Faculty of Health, University Paris Cité, 75006 Paris, France; 5School of Social Work, Sherbrooke University, Sherbrooke, QC J1K 2R, Canada; 6Department of Psychiatry, Bicêtre University Hospital, Assistance Publique Hôpitaux de Paris APHP, Hôpitaux Universitaires Paris Saclay, 94275 Le Kremlin Bicêtre, France; 7CESP, INSERM U1018, Moods Team, Faculté de Médecine Paris Saclay, University Paris-Saclay, 94275 Le Kremlin Bicêtre, France

**Keywords:** male victims, sexual abuse, impact, resilience

## Abstract

The increasing prevalence of sexual abuse calls for exceptional awareness of its multidimensional impact on the mental, sexual, and social wellbeing of male adults. This study aims to deepen the overall understanding of sexual abuse consequences; to highlight some common resilience factors; and to strengthen therapeutic and social support. In this qualitative research, we conducted seven semi-structured interviews with male victims of sexual violence. The data were analysed with the interpretative phenomenological analysis. They shed light on the great suffering linked to sexual violence, and on seven themes which are seemingly pillars of resilience: bond to others, bond to the body, making sense of things, expression, rediscovering oneself, institutions, and finally, learning and commitment. The exploration of these themes reveals several avenues for adjusting care, most of which imply the importance of raising awareness so that spaces receiving the victims’ word can emerge.

## 1. Introduction

The literature shows varying prevalence rates of male sexual abuse, with a tendency toward increase [1]. In the United States of America, the National Intimate Partner and Sexual Violence Survey (NISVS) of 2017 indicated that 30.7% of men report having been victims of sexual violence [2]. Moreover, 16% of male victims would be sexually abused before the age of sixteen [3]. In a recent study including children and adolescents from Barbados and Grenada, the rates of sexual abuse outside and within the family were higher for boys than girls [4]. As for Asia, the estimated prevalence of childhood sexual abuse (CSA) among boys ranged from 1.7% to 49.5% [5]. On a worldwide scale, it would range from 3% to 17% [6]. Despite a significant prevalence, it is possible that male sexual abuse remains highly underestimated, as few men report sexual abuse.

Considering the growing prevalence of male sexual abuse, it is important to investigate its impact on individuals and how they might overcome its negative sequelae. The traumatic impact of sexual abuse seems to be expressed in the context of traumatic dynamics relating to sexuality, interrelationships, powerlessness, and stigmatisation [7]. Emotions tied to sexual trauma can lead to conflicts with identity, as well as masculine norms and stereotypes [8]. Those would include aggression, rejection of “feminine” characteristics, stoicism, preoccupation with sex, being an economic provider, and being the protector of the home and family [3,8].

Easton et al. [9] estimate that it takes an average of twenty years for a man to talk about his history of sexual abuse, due to the stigma induced by current norms of masculinity [10]. This prevents the creation of intimacy [8]. Hudspith et al. [11] add that rape myths may generate feelings of self-blame and shame, which discourage victims, male and female alike, from revealing the abuse. Amongst common gender-based rape myths, one reads “only gay men are raped; heterosexual men are not” [12]. Such false beliefs reinforce secrecy, as well as the stigma induced by norms of masculinity.

Some cultural or religious groups amplify secrecy [13,14]. Findings of a study investigating childhood sexual abuse within German catholic churches indicated that 62% of children were male [13]. In most cases, the relocation of perpetrators hinders the disclosure of sexual abuse and its future prevention [13]. Similarly, a study conducted within the Haredi community suggested that the taboo surrounding sexuality, the fear of sinning, as well as the valorisation of obedience, induced lesser tendencies for individuals, families, and peers to reveal the abuse [14]. Some customs, such as child marriage or child genital mutilation, also put children at risk [15]. Although sexual abuse within religious groups often results in feelings of worthlessness, mistrust, or spiritual struggles, faith may support resilience by providing hope [16].

Regardless of religious background, studies show that the sense of stigma is stronger when the perpetrator was a mother or a woman, as unconsciously, men tend to be more often associated with the status of being the perpetrator of sexual abuse [10,17]. The mistaken belief that women cannot be offenders can discourage recognising an experience as abusive if it involves a woman or a mother [10]. In a study focusing on men who were forced to penetrate women, Weare [18] shows that this form of female-to-male sexual abuse often results in anxiety, depressive and/or suicidal thoughts, self-harm behaviours, mistrust, and negative feelings of anger, shame, and isolation. Being subjected to penetration would also worsen the sense of stigma and feelings of shame [3].

Feelings of powerlessness and loss of control, reported by many male victims, seem to be thought of as being a consequence of male gender socialisation [3,9,17]. Feelings of powerlessness would result from a man’s recollection of being defenceless and would pervade the emotional and relational aspects of his adult life, during which he may feel unable to succeed or to overcome the challenges he struggles with [17]. Additionally, traumatic memory, sometimes referred to as “flashbacks”, seems to focus on bits of the traumatic event and is often unexpected [19]. Thus, traumatic memory might worsen the overall feeling of hurt and losing control, hence the defensive use of avoidance strategies [19]. Traumatic memory figures among numerous symptoms of PTSD, along with persistent stress responses, hyperarousal, dysphoric mood, and circadian rhythm dysregulation [20], and increases the risk of psychoactive substance use and suicidality.

Male victims may also experience great fear in facing contact or aggression from others. Some would tend to defend themselves through the embodiment of a hypermasculine and controlling personality [9,17,21]. Following sexual abuse, many of them show disturbances in identity construction and socialisation, as well as in their sexual identity development [8].

Given the discrepancy between sexual abuse, its aftermath, and beliefs about what a man is supposed to be, some authors point to the need for male victims to renegotiate masculine norms [8,17]. The renegotiation of these norms in adult victims would be a first step towards the “breaking of the victim–offender cycle” [8,22]. However, assessing whether a victim manages to break the cycle of violence or not misleads us into believing that reproducing the abuse is unavoidable. Actually, though perpetrators of sexual abuse often endured it themselves, few men who endured sexual abuse do reproduce what they went through.

Emotional reassurance, social support from friends and family members, and positive disclosure experiences are also described as being of major importance in the resilience process [10,23]. In addition, the #MeToo movement on Twitter is therapeutic for many victims, providing a space for emotional support, sharing, perspective, community, and awareness, as well as perpetrator and general population accountability [24].

Few studies investigate male resilience, especially within a French context. In France, approximately 16.1% of men report having been victims of rape or attempted rape [25]. Nevertheless, male sexual abuse is known to be widely underreported. Childhood sexual abuse is reported to be widespread among family, and literature finds this to be the case in the Catholic Church too, where a culture of secrecy is omnipresent and gender norms foster violence against women and children [26]. Although changes in French legislation slowly allow for better recognition of rape and sexual assault, the taboo surrounding male sexual abuse remains strong. In addition, professionals lack information regarding the care of victims [25]. Therefore, to pursue reflections on resilience, this research will attempt to describe the impact of sexual abuse on the lives of men who endured it, and to identify other factors promoting resilience. Its objectives are: (a) to deepen the overall understanding of male sexual abuse consequences; (b) to highlight common factors of resilience; and (c) to strengthen the help and therapeutic care offered.

## 2. Methodology

### 2.1. Sample

Seven men, aged between 26 and 54, participated in the study. Selection criteria consisted of identifying as a man and having endured sexual abuse during childhood. Since resilience is seen as a subjective experience to be explored, it was up to each individual to define it.

The participants took part in the study voluntarily. To recruit participants, several posters and pamphlets, outlining the objectives and implications of this research, were sent to associations assisting victims of sexual abuse. Initial discussions were held with interested men by telephone, during which the study course and each person’s expectations were clarified. No participant withdrew.

This study was approved by the University of Lorraine and is registered on the research register with the number 2021-194-1. The study was performed in accordance with the Helsinki Declaration. Each participant completed and signed a consent form detailing the context in which their data would be used and stipulating their rights of withdrawal and refusal.

### 2.2. Data Collection

The participants’ accounts were collected using semi-structured interviews. The interviews’ semi-directed nature was intended to encourage the participants’ word and associativity, as “humans are meaning-making organisms” [27].

The interview consisted of fourteen open questions. The first part aimed to build trust and explored each participant’s personal history. It focused on their perceptions of male sexual abuse understanding in our society, as well as their own experiences of such violence, and dealing with legal proceedings or receiving psychological support. The second part aimed to provide an in-depth description of the abuse consequences. Finally, the last part focused on the participants’ resilience and their suggestions for therapeutic and social care improvement.

The interviews were held individually between April and June 2021. Due to distance and the COVID-19 pandemic, six of the interviews were conducted via video conference and one of them via a telephone call. The video conference interviews were recorded and the telephone interview was transcribed. All were then transcribed and submitted to the participants so that they could revise some of their answers as a final validation. This paper was written within a French context.

### 2.3. Data Analysis

Data were analysed using the Interpretative Phenomenological Analysis (IPA) method, defined by Antoine [28] as an attempt to “apprehend what the experience of being is and what constitutes the lived world”. This involves placing individuals and their subjective perceptions at the heart of their experiences [29]. The term ‘phenomenology’ is composed of the Greek words referring to ‘phenomenon’, and ‘logos’, meaning ‘discourse’ [28]. In other words, phenomenology is interested in the meaning a person attributes to an experience. It intends to put a singular phenomenon into words through the interpretive work of the analysts [27].

In this study, putting the participants’ accounts into perspective has allowed us to emphasise the strengths and difficulties of their path after the abuse occurred. Their accounts were then organised around various sub-themes. Verbatims and clinical theory aimed to illustrate the deduced resilience factors as well as their therapeutic care suggestions. According to IPA, the combination of experiences studied one by one leads to a collective experience from which it is possible to draw general hypotheses [27].

The application of this methodology and construction of the protocol were thought out in relation to the Coreq-32 method: Consolidated Criteria for Reporting Qualitative Research, consisting of a 32 items list intended to guide the analysts in their research [30].

### 2.4. Reliability of Results

In order to increase data saturation and exhaustiveness of the deduced themes, the scoring grid and results were subjected to several revisions by experienced researchers. The themes were designed in the light of the participants’ accounts and were revised in the light of subsequent reflections. Finally, the article was sent to all participants so that they could share their own perceptions, along with the transcripts.

## 3. Results

In a few words, resilience is about how a person copes with their pain. In this study, it is more specifically about how the participants managed to overcome sexual abuse negative sequelae. The analysis of the participants’ accounts (Table 1) produced two main themes, one of which was the impact of sexual abuse and the other was resilience. The impact of male sexual abuse included psychiatric implications, sexuality, self-esteem, relationships with others, and self-fulfilment. Psychiatric implications, sexuality, and self-esteem primarily pertained to the individual, while relationships with others and self-fulfilment appeared to be directly influenced by the surrounding environment. As for resilience, discourses revealed seven dimensions: bond to others; bond to the body; expression; making sense of things; rediscovering oneself; institutions; and finally, learning and commitment. Figure 1 summarises the findings.

### 3.1. Impact

#### 3.1.1. Psychiatric Implications

Psychiatric implications encompass various psychiatric symptoms experienced by the participants. Many participants reported post-traumatic stress disorder (PTSD) symptoms, depressive symptoms, and suicidal thoughts. Some may also have experienced psychosomatic pain, such as headaches or back pain. Others experienced very fluctuating emotions: “*It was really… a rollercoaster. Emotionally. That is to say that… one minute I was feeling very, very bad, without any external reason, there was no real external reason. And one minute I was very, very well*” (Participant 1). Some participants also mentioned obsessive thoughts about the trauma they had experienced: “*I can’t go a day, in fact, without saying the word “rape” in my head*” (Participant 2).

#### 3.1.2. Sexuality

Sexuality refers to the participants’ challenges with intimacy and sexual thoughts. Some express fears of causing harm or abuse, therefore restraining from engaging in relationships: “*To feel like a threat to the child, as if we were replaying the same scenario. It’s living with the thoughts of your abuser*” (Participant 3). Some feel intense pressure; they are scared to do to others what was done to them, regardless of their strong intent not to harm anyone: “*I have this pressure. I have three generations of men before me who succumbed to violence, I am the fourth, and I have the responsibility not to repeat what happened before.*” (Participant 2).

Some experience difficulties in letting go and feeling vulnerable, which notably occurs alongside sexual difficulties: “*I know that in terms of sexuality, which is the biggest symptom of the thing, really… I find it very difficult […] to go and feel vulnerable, that is, to let go. […] It’s a reaction of my body, really*” (Participant 4).

#### 3.1.3. Self-Esteem

Self-esteem highlights how the abuse negatively affected the participants’ self-image and self-love. The participants spoke extensively about the impact of the abuse on their feeling of self-worth and confidence. They report a significant wound, taking the form of an “*emotional void*” (Participant 2); of a bruised relationship with their body or a feeling of no longer belonging among men: “*All that I had… built up as an image of myself, in any case, in terms of… as a weak person… in any case, an inferior being*” (Participant 1). This degraded self-esteem could explain behaviours of neglect that some of them mentioned, and which corroborates the depressive dimension mentioned above: “*I neglected myself. In fact, I didn’t take baths anymore, I didn’t shower any more, I didn’t wash myself anymore*” (Participant 6).

Heavy feelings of shame and guilt contributed to aggravating this lack of self-love: “I’m proud of what I’ve become, but I’m ashamed of what I was. I feel like I was some kind of… some kind of wreck that couldn’t protect himself. […] I am soiled” (Participant 2). More precisely, some defined those feelings as being of defilement, humiliation, or dishonour: “For a very, very, very long time, I said to myself “but it’s a disgrace for me to have lived through that, to have been weak and to have… to have been unable to prevent it”. […] I felt humiliated to have experienced that” (Participant 1).

#### 3.1.4. Relationships with Others

Relationships with others encompass emotional obstacles that complicated the participants’ abilities to form meaningful connections. It is reasonable to think that such negative feelings influence the withdrawal tendencies expressed by some participants, such as difficulties in verbalising and expressing emotions: “*I found it difficult to express my feelings, my emotions, or what I was experiencing*” (Participant 5). In addition, several of them noted the fragility of their personal limits: “*It was also a time when I didn’t dare to say no. I didn’t like to say no because I thought that saying no would hurt the other person. It was negative, it wasn’t nice, and so I felt obliged to say yes*” (Participant 1).

Some participants mention a need for love, support and recognition that is sometimes intrusive and invasive: “*There is a pitfall, however, and it’s… since it’s an abyss inside, it’s to absorb the other. It’s being so in demand that it can be suffocating for the other. It’s even giving the impression that you are deliberately not accepting their help*” (Participant 7). This can lead them to conflict-avoidance and self-effacement behaviours: “*I had an… enormous need for recognition. And I also realised that in all my social relationships, I always tried to be liked. In fact, I had a need to be loved that was enormous. […] I never made any waves. In fact, I rarely took positions, or I always tried to avoid conflict and what can be a bit itchy for people. […] I always did more for others than for myself, I completely forgot myself*” (Participant 5). Others, on the other hand, displayed asocial behaviours: “*The fact of really being elsewhere, of floating, […] of not really being in your body and all that. It also adds… yeah, a bit of anti-social behaviour, the feeling of being out of place, of being misunderstood*” (Participant 6).

They also mentioned difficulties in building friendships with men, which went hand in hand with a complex relationship with masculinity: “*It’s good to see men again, because it allows me to reconcile with masculine gender*” (Participant 2). Finally, many find it difficult to build harmonious and/or lasting emotional relationships, underlining an emotional instability, which goes hand in hand with a feeling of isolation and abandonment: “*I can’t manage to have stable relationships, because I don’t feel stable in my body, and I feel even less stable when I’m in a relationship with someone else’s body*” (Participant 3); “*It’s a state of abandonment, of not having any attachments, and to feel detached from everything is hard*” (Participant 7).

It seems important to note that aggressors were all known by their victims. It is reasonable to assume that experiencing abuse at the hands of trusted peers or family members could induce a profound feeling of betrayal and insecurity. This, in turn, might complicate the building of future, sincere relationships by creating fears, defensive strategies, and long-lasting distrust.

#### 3.1.5. Self-Fulfilment

Self-fulfilment is characterised by the participants’ struggles to find a satisfactory place in contemporary society. Resulting from an inability to project oneself into the future, schooling has been difficult for some, and career paths are fraught with breakdowns. Participant 1 mentions self-destructive behaviours that may have, at some point, prevented him from fulfilling his potential: “*For me, one thing that really… put up barriers to getting better, was my drinking. It was like a cycle. I drank to forget… and it was every day. It was like a routine*”.

With regard to feelings of anger and injustice, it would seem that anger directed at the aggressor subsided over time, unlike anger directed at society, which tends to remain unchanged: “*I still have a kind of anger that is always present. Against institutions, against justice, actually. Against French laws which are really… not even badly made, actually, but which are made against the victims, really*” (Participant 6). Four participants engaged in judicial proceedings. At the time of the study, only two aggressors were convicted; one aggressor was not convicted; and one case was dismissed. Some participants may have relied on society to recognise their painful experiences and hold the aggressor accountable. Being confronted to different outcomes may reinforce feelings of injustice, anger, stigmatisation, and isolation.

Thus, the abuse impacts their overall well-being in society, which they do not perceive as safe. It could make it difficult for them to feel like this is a place where they could thrive.

### 3.2. Resilience

#### 3.2.1. Bond to Others

Bond to others represents the participants’ capacity to establish supportive interpersonal relationships. According to Participant 4, “*resilience comes from others*”. Entourage, support, and the feeling of being connected to others are the most cited requisites for resilience, although this bond remains impacted by the abuse. The accounts highlight a need to create bonds and the associated difficulty. The latter often relates to emotional needs that are considered too intense, to a fear of rejection, or to a distrust in human relationships: “*I have the impression that it scares me, and then it’s fixed and it’s over, once things are settled*” (Participant 7). However, this is not a systematic obstacle to the reconstruction of meaningful relationships.

For Participant 1, getting better would be a matter of having found the right people to lean on: “*in my opinion, it was more the outside that got me through than the inside*”. The interviewed men reported a greater number of positive disclosures and support following those disclosures. For them, positive disclosures were a first experience of being recognised and supported: “*there was a “before” and an “after”. In the after, I had the feeling of being really understood*” (Participant 6); “*It was liberating*” (Participant 5); “*Piecing myself back together was a bit easier because… well, my kind environment helped a lot, that’s for sure. There was no shaming, no question*” (Participant 4).

The accumulation of positive experiences makes participants feel safe and encourages them to talk about the abuse. In turn, it helps them to break out of their isolation, to elaborate, and to work on themselves: “*I find it very touching, and to make relationships that are much deeper and more meaningful than just staying on the surface and talking about good weather*” (Participant 4). Being able to share their experiences fosters the creation of transparent and authentic relationships: “*A better understanding of what I am going through, a better interaction with who I really am, and not with an image of myself. To have authentic, honest relationships, based on solidarity, based on mutual understanding*” (Participant 3).

Furthermore, many of the participants belong to a support group and have identified this membership as a strong lever of resilience. Being a part of a social space where their experience is not rejected allows them to reconnect with others, thanks to the absence of stigmatisation, which can otherwise lead to a feeling of abnormality: “*to actually belong, to be, to be there, to be human, to be normal*” (Participant 6). Support groups have the function of bringing people with similar experiences together, which allows its members to identify with each other and to engage in a common reflective process at the service of each individual: “*the fact of being understood directly by people who have experienced the same thing does a lot of good, actually. And every time, we are there to talk about very painful subjects, but something positive always comes out of it*” (Participant 5). For Participant 3, it is a way to feel less alone and to understand himself better: “*To externalise what I am going through, to talk about it, to write about it, to make videos, to talk about it with others, to create mutual aid networks… to not feel all alone in this. […] And in a way… I am not just helping others; others are also helping me*”.

#### 3.2.2. Bond to the Body

Bond to the body denotes how participants invested in and connected with their own bodies. In this study, participants often struggled with connecting with their bodies. As such, working on that connexion seems to be one of the fundamentals of resilience. It is often, in this study, associated with meditation and relaxation practices such as conscious perceptions of soothing stimuli (ASMR: Autonomous Sensory Meridian Response), breathing exercises, yoga, mindfulness, etc. Specifically, meditation would be a means of mitigating dissociation behaviours by “*re-establishing a connection with oneself*” (Participant 3) and by experiencing their own needs and limits: “*I’ve taken refuge in my head, and… meditation, in particular, serves me. These activities serve me to try to… to reconnect with my body and my sensations*” (Participant 2).

Reclaiming their body can be a way of regaining control and gaining confidence: “*I needed to… […], to do something with my body, to prove to myself that there were still things I could do*” (Participant 4, in relation to climbing). Some also stressed the feeling of unity and wholeness: “*To be able to feel like a unit, like an individual, an individual in the sense of a functioning unit, it’s immensely difficult. […] It’s as if I were several and at the same time I was one*” (Participant 3). This can be achieved through psycho-corporal re-education: for instance, several men participated in fencing workshops. Learning to love their body again and understand their body’s instincts are two other components of resilience. For Participant 4, this notably concerns reflecting on unexpected reactions when feeling in danger: “*It happened once or twice, I don’t know… my ex was bothering me, something like that, and… like, I couldn’t see her, and she started stroking my nipples, and I slapped her. […] We stared at each other. We were completely speechless, not knowing nor understanding what had happened*”.

#### 3.2.3. Expression

Expression encompasses the various means by which the participants were able to communicate or shape their stories. Participant 7’s experience shows that talking about the abuse is not enough to get better: “*In a day, I can have moments when I feel good somewhere, but the benefit would be to build. That is to say that these moments when I feel good remain in the background to carry me through the following days. But there is no such thing. […] I think you’ve put your finger on it: the mistrust. But it’s kind of a bad reflex*”. In spite of everything, words remain central for all participants.

Spaces for expression play an important part in creating bonds and breaking down isolation. Participant 3 identifies the production of his autobiographical video as a turning point in his path of self-reconstruction: “*The video had an effect of getting people to recognise what I’ve been through, to help other people recognise what they’ve been through*”. As for Participant 1 and Participant 6, they share the powerful disclosure of their story on Facebook: “*When the #MeToo movement started, I posted a text on Facebook to say that I had endured sexual abuse. And the same evening, my twin brother sent me a message on Facebook. […] And he was very supportive. Unconditionally.*” (Participant 1); “*I had written […] a long post on Facebook saying that I had been sexually abused, and I actually got a wave… really, a tsunami of sympathy. […] I really felt loved, protected, surrounded. It was a fabulous feeling, it was really… a wonderful moment of my life*” (Participant 6). In addition to creating supportive networks, artistic expression supports the transformation of painful experiences as well as the externalisation of emotions and personal experiences. Participant 2, for example, writes poetry: “*I have a notebook in which I write down all my feelings*”.

Music, whether listened to or produced, can be a means of regulating emotions. Steady practice of sport, specific to many of the participants, allows them to evacuate, to escape, and to regain control over the body while participating in an overall feeling of well-being. Participant 6, through his account of a “psycho-magical act”, recounts how spirituality healed a deep part of his soul: “*It was really incredible. Because there’s a sort of cry, a literal and symbolic cry in me, actually, that… came out*”. All those elements emphasise the importance for men to be able to express their emotions, their difficulties, as well as their hopes.

#### 3.2.4. Making Sense of Things

Making sense of things encompasses the participants’ interpretation of their life stories and their perceptions of themselves within those narratives. Referring to Grossman et al.’s [31] approach to mentalization, “*meaning through action*” would involve altruism and the use of creative expression to elaborate trauma. As shown by the participants, creative expression (books, videos, poems, etc.) often aims to transform pain while putting it at the service of others: “*That it has not been in vain, and that it can… there you go, prevent that from happening to other people*” (Participant 5). Many participants want to reach out to other victims, due to the development of great empathy towards their suffering and a greater tendency to create solidarity bonds.

Other participants make sense of their stories through the “*use of cognitive strategies*” [31]. For Participants 1, 2, 3, and 5, it was a question of understanding how the abuse came to exist. They could resort to environmental, historical, or socio-cultural explanations. In particular, “*patriarchal norms of our society*”, as they quoted, are often pointed out as the source of much violence: “*But that’s why I come back to the boys who did that to me. I think they were completely in it: to exist through the reproduction of what society, their environment, or whatever, gave them*” (Participant 5). Some have a more psychological conception of violence, trying to understand what in the life or personality of their aggressor pushed them to act this way. Participant 4 and Participant 2 talk about the reproduction of traumas experienced by their abuser—a transmission of violence to which the complex dynamic of secrecy is added: “*you can see that when people talk to each other… it’s in the unspoken that dirt happens*” (Participant 7). This understanding enabled Participant 1 to partly forgive his abuser. He describes that as a high point in his path: “*You reach a point where there are several choices. There were two choices: there was a choice where I could get better, a choice where I could get […] worse. […] There’s still a part of me that doesn’t forgive, but it’s… I stopped seeing him as the devil. And that, in itself, has helped me a lot*”.

#### 3.2.5. Rediscovering Oneself

Rediscovering oneself involves self-understanding and adjustment. Experiences of sexual abuse are profoundly destroying for those who endure it. They disrupt a person’s personality and sense of self–hence the importance of rediscovering oneself.

Introspection, counselling, or exploration of childhood elements are cited by participants as some actions leading to a better understanding of the self and various behaviours. For Participant 2, knowing himself better equals “*a greater state of awareness*” and “*greater potential for change*”. This in turn seems to encourage the projection of the self into the future, as well as reinvestment in passions or projects. It also seems important “*to assume one’s life and values without justifying them by the past*” (Participant 2). In other words, distancing themselves from a “victim status” is a way of reclaiming themselves: “*Thanks to support group, I also became aware of the power of words. And to say “I was a victim of, I am a victim of”, that’s rubbish! I mean, being a victim is a lousy status […], it’s like you’re just going through life! Say “I’ve been attacked”, that’s it. It’s someone other than I. I didn’t ask for anything, and someone else came and did that. […] The person who has suffered does not have to be responsible*” (Participant 4).

Rediscovering themselves involves many other things: learning to understand themselves well enough to be able to adapt to life’s situations (“*Understanding how our past influences who we are today… […], I think it’s extremely helpful in order to cope better. Understanding our own emotions and freeing from them is a strength”* Participant 5); relearning to set limits and listen to themselves (“*Until now, I didn’t actually listen to myself. I did things according to others, and not according to me and my limits*” Participant 5); and finding inner peace and serenity.

#### 3.2.6. Institutions

Institutions play a role in providing social recognition of the participants’ suffering. Whether they are judicial, medico-social, or associative, institutions play a primordial role in legitimising the pain of people who have experienced sexual abuse. In a context where male sexual abuse remains taboo, they place a man in a group where his experiences are accepted, which helps to validate the events, as well as the perceptions and emotions that result from them: “*The first time I went there, this notion of understanding was basic. We are all the same, in a way. […] Resilience comes from others. It’s a bit sad, because when you’re in a society that constantly tells you that it’s your fault you’ve been raped, well, it’s complicated*” (Participant 4). For Participant 5, hearing, in his support group, that what he had been through was violent helped him to realise the extent of the abuse he had suffered, and to not minimise or trivialise it: “*sometimes you wonder whether you have experienced them or not, it’s complicated. You think that in the end, it’s not much compared to other stories*”.

Acceptance, according to Participant 2, can lead a man to feel authorised to represent himself as a legitimate victim of the abuse: “*First of all, to allow victims to become aware of themselves, to be able to authorise themselves to say they are victims. Because being confined to the role of the aggressor does not allow one to consider oneself a victim. It is difficult to do so*”. Then, this would allow him to move from a before to an after. Turning the page becomes possible. For instance, it could help place the blame on the aggressor: “*I regret not going to the trial. I didn’t hear the verdict, I didn’t hear the judge say that he’s guilty, or that he’s responsible for his actions, and… I even think he would have said a word, or that he would have apologised for everything*” (Participant 4).

In addition to these functions of recognition and legitimisation, institutions also play a considerable role in breaking down isolation and in the social reintegration of male victims—through the building of reference points and networks, in particular.

#### 3.2.7. Learning and Commitment

Learning and commitment emphasise the importance for participants to understand themselves in order to better engage in collective experiences. Learning and commitment are two levers of resilience that may intertwine. Learning about the mechanisms of violence, the sequelae of violence, and the sociological conceptions of violence would not only allow male victims to understand a personal and collective experience but would also help them to develop adapted defence strategies. Understanding uncovers tools that can be used as a starting point to reflect on oneself. It is reasonable to assume that the active dimension of learning counteracts the feeling of helplessness, which arises from difficulties and suffering that the participants do not always understand until they establish a direct link between their experiences and their problems.

If learning leads to greater self-understanding and self-indulgence, it can be assumed that it builds confidence and self-esteem. Thus, some participants plan to share their knowledge with others, especially with the media, in order to get others to understand as well. Thereby, learning supports a process of commitment. Whether it is towards others or towards change, commitment seems to give purpose to the suffering endured, as well as strength to bear it: “*I just create a space where people can actually express themselves, and are listened to, and feel like they are not alone*” (Participant 6).

What emerges from the collected accounts is a deep desire to break the silence, and “*turn it into a desire to say it*” (Participant 5).

## 4. Discussion

This study aimed to deepen the overall understanding of sexual abuse consequences and to identify common resilience factors, in order to strengthen therapeutic and social care. The collected accounts highlighted the painful and lasting impact of sexual abuse. Feeling like the memory of the abuse remains on the verge of coming back at any time corresponds to what Brewin [19] describes as traumatic memory. It arises in a raw and intrusive way and plunges a person into the sensation of reliving the violent event. According to the study, it occurs during stress-inducing situations and generates great emotional distress [19]. Traumatic memory punctuates the resilience process by causing ruptures that can make a victim feel as though they are returning to the starting point. In this sense, the process of resilience remains in a constant state of flux.

The accounts also highlighted seven dimensions of the resilience process. While all participants indicated that sexual abuse had a particularly strong impact on the quality of their connection to others, the ability to rebuild trust and build supportive relationships was identified as an important step in the resilience process, especially seeing as building trust might have been deeply challenging for those who were abused by close friends or relatives. These possibilities seem to be associated with more positive disclosure experiences. This is consistent with an observation by Gagnier and Collin-Vézina [10], who explain that social support facilitates talking and integration of the narrative into a subjective story, which is essential in trauma resolution. Being able to share these experiences fosters transparent and authentic bonds, which have been shown by Kia-Keating et al. [8] to be an important resilience factor. Endorsing those findings, Roberg et al. [32] found that the emotional openness and a reduction in shame could stem from trust, recognition, and acceptance often provided by other victims in support groups.

One of the characteristics of dissociation, defined by the American Psychological Association (1994) as “*a disorder or alteration of identity, memory, or consciousness*”, is an emotional, memory, or identity detachment. Sexual abuse affects a person in their flesh; thus, this fragmentation of the self may involve an absence of bodily sensations. Reinvestment and re-engagement of the body have therefore been emphasised as major issues in the self-reconstruction of the participants. They needed to be able to experience, protect, and reconnect with their internal world.

Concerning relationships with the self and with others, speaking and sharing experiences are two key elements. Whether in a personal, relational, therapeutic, or counselling context, verbal or artistic expression is at the heart of resilience. It encourages the verbalisation and concretization of emotions and thoughts and transforms a meaningless experience into words that will echo the experience of others. In fact, it breaks down isolation. In this particular study, it facilitated the reconstruction of perceptions that refuted social prejudices and were in line with the participants’ internal experiences.

Ultimately, expression and transformation guide victims towards the possibility of making sense of their experiences. Sense-making, according to Grossman et al. [31], has to do with the mentalization of the traumatic event because it allows an otherwise meaningless experience to be understood and integrated. Integrating it facilitates acceptance in order to move on. For some participants, it was important to be able to understand what, in their social environment or in their abuser’s personality, had led to the abuse. For others, reaching out to others was a way to give purpose to their suffering. Among the protective factors identified by Lighezzolo and De Tychey [33], altruism would be a way of soothing our own wounds by ‘*giving happiness*’.

Furthermore, it seems that sexual abuse breaks down a person’s sense of unity and continuity. In fact, the participants emphasised the importance of learning to understand themselves better. Overall, they believe that working on themselves improves their self-esteem. Knowing that they can do good for others, being aware of what they have overcome, and having worked on themselves for a long time are cited as the main reasons for improved self-esteem. According to Roberg et al. [32], insight would be a means of reducing shame and taking back control. In addition, the deconstruction of a damaged self-image could help men to let go of the guilt they feel in relation to the events, and to show more kindness towards themselves.

Learning fosters self-discovery and commitment. All participants spent a lot of time enriching their thoughts on the subject of sexual abuse. The result is a strong commitment, which manifests in the form of help proposals and general suggestions supporting the social and therapeutic implications of this study.

On the one hand, participants attach particular importance to the awareness and verbalisation of sexual abuse among the general population, as well as among certain professionals. Indeed, those in the latter would benefit from being trained in receiving testimonies or providing care for male victims. Hudspith et al. [11], for instance, propose several suggestions to reduce rape-myth-related beliefs amongst jurors. In order to encourage the majority to break the taboo, and to effectively support victims, it is important to maximise the visibility of associations or institutions specialising in sexual abuse victims’ assistance. Furthermore, professionals, facilities, and organizations able to receive word of male victims would benefit from diversification (such as referents in schools or companies, support groups, therapeutic workshops, and psychological sessions accessible to all).

Research, prevention campaigns, testimonies, social networks, media, debates, arts, and education play a key role in making citizens aware of the effects of sexual abuse on others and in helping victims to grasp the reality and/or severity of the endured violence. Greater representation of sexual violence against men would reduce the stigma inherent in norms of masculinity: for example, through inclusive communication, meaningful film portrayals, and collective accountability. It is important to place the blame on the perpetrator rather than the victim, regardless of their gender. Similarly, citizens would benefit from improved laws on the protection of victims (enforcement, creation of new laws, financial compensation, age of consent, statute of limitations, etc.), and the treatment of perpetrators.

Finally, some participants mentioned the need to educate young people about sexuality, including notions of consent, violence, and physiology (e.g., pleasure, orgasm), in order to reduce the feelings of guilt and shame attached to sexual abuse. Similarly, the integration of a more flexible and harmonious male ideal, in which young boys are allowed to express their emotions, seems fundamental to preventing silence and isolation.

Some of the reasons why a male victim of sexual abuse may not talk include the feeling of stigmatisation induced by those norms, as well as the fear of being blamed or rejected [34]. It thus remains a sensitive topic, if not a taboo, amongst society. Studies show that men may suffer from many of the same difficulties as women do: sexual dysfunctions, damaged self-image, feelings of shame and guilt, depression, anxiety, feeling of isolation, and relationship-building difficulties [7,35]. As the current study focused only on men, further research should explore the specific impact of gender-based sexual abuse on men’s image of their own masculinity.

Nevertheless, with reference to male gender socialisation, current understandings of masculinity complicate their own comprehension and acceptance of the abuse and related emotions. According to Weare [18], “*even if some negative impacts are recognised, the gendered construction of masculinity allows for the effects on men to be minimised*”. Hence, this research means to report on male sexual abuse impact and tries to propose suggestions to improve their mental health.

## 5. Limitations

A limitation of this study is the small number of participants. Few male victims of sexual abuse report as adults, which limits the possibility of recruiting more participants. Also, the study used a non-random sample and, therefore, the sample may not be representative of all sexually abused men. In addition, this study highlights findings mostly related to the French context, so international studies should be conducted to assess differences related to country contexts. Finally, resilience is a personal process; as such, it may differ for younger people, as well as people who were more recently abused. Moreover, men who agreed to participate may not have experienced resilience the same way as those who did not participate.

## 6. Conclusions

This research aimed to describe the impact of sexual abuse on the lives of men, and to identify processes of resilience. Phenomenological analysis of the narratives enabled us to identify various recurring themes in the participants’ discourses. The bond to others stresses the importance of being able to rely on someone and to create supportive relationships that generate verbalization. In a family or institutional setting, this bond must allow the person to share his story. Sharing or, in other words, expression, is a second key lever of resilience. It can take the form of speaking, physical engagement or artistic creation… but in all its forms, expression is often the starting point of elaboration, which in turn can lead to meaning-making. Similarly, it seems important to accompany victims in a process of discovery that matches their own need to understand their suffering, and thereby understand themselves better. If engagement could be a source of resilience for the people we met, it reflected their need to commit to those who need it—and the need for society to commit to victims.

## Figures and Tables

**Figure 1 healthcare-11-01868-f001:**
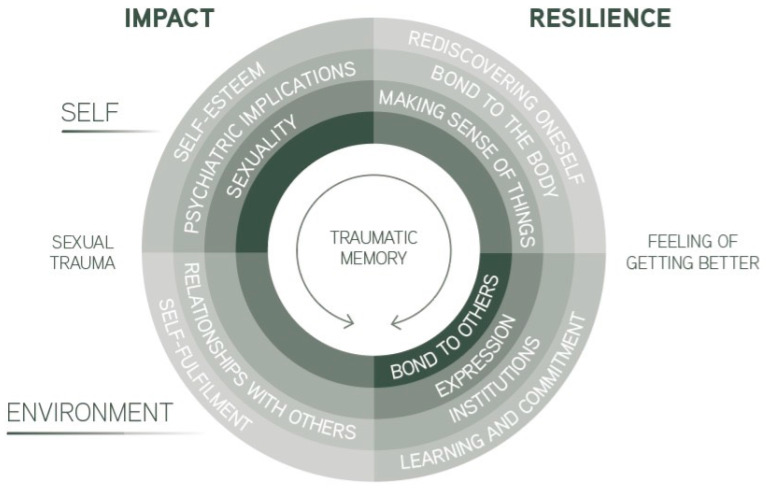
A scheme illustrating the dimensions of the impact of male sexual abuse and aspects of resilience among victims.

**Table 1 healthcare-11-01868-t001:** Sociodemographic, interview, and abuse-related characteristics of the sample.

P	Age	Profession	Nature of the Abuse	Age at First Violence and Occurrence	Aggressor	Psychological Support	Judicial Proceedings	Duration of Interview *
1	26	Employee	Sexual touching, rape	7 years old—for a year and a half	Classmate (male)	Yes	Yes—not convicted	1:48:43
2	28	Student, employee	Rape	8 years old—several times	Half-brother	Yes	Yes—convicted	5:10:30
3	47	Employee	Sexual touching, rape	8 years old—for 12 years	Father	Yes	No	1:18:59
4	26	Employee	Sexual touching	9-10 years old—for 9 months	Friend of family (male)	Yes	Yes—convicted	2:09:04
5	43	Employee	Rape	12–13 years old—several times	Classmates (males)	Yes	No	1:59:47
6	35	Employee	Sexual touching, rape	9 years old—for 10 days	Half-brother	Yes	Yes—dismissed	1:30:24
7	54	Unemployed	Sexual touching, fellatio	4 years old—for 15 years	Uncle	Yes	No	3:15:00

* The interviews lasted between one and five hours. These length differences can be explained by some participants’ greater need to reflect on questions and to detail their experience.

## Data Availability

The datasets used and/or analysed during the current study are available from the corresponding author upon reasonable request.

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
