# Peer review of "Male Victims of Sexual Abuse: Impact and Resilience Processes, a Qualitative Study"

_healthcare, 2023, doi:10.3390/healthcare11131868_

Round 1

Reviewer 1 Report

An interesting paper that shares and gives voice to 10 men's lived experiences of sexual abuse and their recovery/resilience. Male sexual abuse is a topic that needs much more discussion than currently and the authors are applauded for their study and paper which brings this further into light. A few notes:

Abstract

- Please include a brief description of the methodology used

Introduction

- It would be beneficial to begin with a broader introduction on sexual abuse before discussing male sexual abuse

- Line 83: Insert "and" before “c) strengthen…”

Data collection

- The terms interview, questionnaire, and survey seem to be used interchangeably. What specifically was the methodology used?

- A statement of where the study was conducted would be useful. It was not clear until the end of the paper that this was within a French context.

Data analysis

- Line 124: Incomplete sentence

Results

- Line 142: Capitalise Table 1

- Line 148: "Drawing 1" Should this be Figure 1 instead?

- It would be useful if the table and figure were inserted after their first mention as opposed to further along the paper

3.1 Impact

- This paragraph discusses the 2 themes of impact and resilience broadly as opposed to specifically the theme of Impact. Perhaps this paragraph would be better merged within the previous section.

- A brief description of each dimension within the theme of Impact would be appreciated (similar to the description given about the dimensions of Resilience)

Discussion

- Line 442: This is a very interesting point and it would be great if this could be first introduced and discussed further within the introduction. Especially as Drawing 1 touches on this concept.

- A more critical discussion around the gendered implications of sexual abuse would further add to the impact of this paper. For example, discussion of masculine gender norms, taboo and stigmatisation of male sexual abuse, etc would provide this paper with more novelty as men's experiences of sexual abuse as opposed to sexual abuse as a general topic.

- The authors do well in engaging with literature to support the recovery/resilience process of these men's stories. However, it would be great to see more discussion of how the results of this study compare (either in support of or contradiction) to other studies.

Overall, a very interesting paper that sheds light and gives voice to men's lived experiences of sexual abuse (a taboo/stigmatised issue).

Author Response

We would like to thank the reviewer for his/her help to improve our manuscript. We are very grateful. For ease of review, changes have been in green in the text.

Reviewer 1: An interesting paper that shares and gives voice to 10 men's lived experiences of sexual abuse and their recovery/resilience. Male sexual abuse is a topic that needs much more discussion than currently and the authors are applauded for their study and paper which brings this further into light.

Response: We thank the reviewer for his/her comment.

 A few notes:

  1. Abstract

- Please include a brief description of the methodology used

Response: We thank the reviewer for his/her suggestion. We completed the abstract. The following sentence was added: “In this qualitative research, we conducted 7 semi-structured interviews with male victims of sexual violence. The data was analyzed with the interpretative phenomenological analysis.”.

  1. Introduction

- It would be beneficial to begin with a broader introduction on sexual abuse before discussing male sexual abuse

Response: We thank the reviewer for his/her suggestion. We have added the following paragraph: “Over the last decade, some authors have highlighted an increase in childhood sexual abuse. Considering that growing prevalence, it seems important to be aware of the impact of sexual abuse on an individual and to wonder how they might overcome it. Despite the fact that sexual abuse against women is becoming more widely discussed, sexual abuse against men tends to remain a taboo”.

The following references were added into the section: Blanchouin and al (2005); Katz and Kamama (2013).

  1. - Line 83: Insert "and" before “c) strengthen…”

We thank the reviewer for his/her vigilance. We completed the sentence.

  1. Data collection

- The terms interview, questionnaire, and survey seem to be used interchangeably. What specifically was the methodology used?

We thank the reviewer for his/her remark. We used the word “interview” for our methodology to be more precise.

  1. - A statement of where the study was conducted would be useful. It was not clear until the end of the paper that this was within a French context.

Response: We thank the reviewer for his/her remark. The following sentence was added in the “methodology” section: “This paper was written within a French context”.

  1. Data analysis

- Line 124: Incomplete sentence

We thank the reviewer for his/her vigilance. We completed the sentence.

  1. - Results

- Line 142: Capitalise Table 1.

Response: We thank the reviewer for his/her remark. We capitalized the line.

  1. Line 148: "Drawing 1" Should this be Figure 1 instead?

Response: We thank the reviewer for his/her remark. We changed “drawing” into “figure”.

  1. It would be useful if the table and figure were inserted after their first mention as opposed to further along the paper

Response: We thank the reviewer for his/her suggestion. We have made the necessary changes.

  1. Impact

- This paragraph discusses the 2 themes of impact and resilience broadly as opposed to specifically the theme of Impact. Perhaps this paragraph would be better merged within the previous section.

Response: We thank the reviewer for his/her suggestion. We merged the paragraph within the “2. Results” section.

  1. A brief description of each dimension within the theme of Impact would be appreciated (similar to the description given about the dimensions of Resilience)

Response: We thank the reviewer for his/her comment. Each dimensions of impact were more specifically defined: “In response to male sexual abuse, discourses revealed five dimensions of impact.

Psychiatric implications, reporting different psychiatric symptoms.

Sexuality, relating to the participants’ difficulties regarding intimacy and sexual thoughts.

Self-esteem, mentioning how the abuse damaged their self-image and feeling of self-love.

Relationships with others, referring to different emotional hindrances which complicated their building of meaningful relationships.

Self-fulfilment, globally characterised by their difficult place in current society.”

  1. Discussion

- Line 442: This is a very interesting point and it would be great if this could be first introduced and discussed further within the introduction. Especially as Drawing 1 touches on this concept.

Response: We thank the reviewer for his/her comment. We introduced the concept of traumatic memory in the introduction section: “Additionally, traumatic memory, sometimes referred to as “flashbacks”, seems to focus on bits of the traumatic event and is often unexpected (Brewin, 2018). Thus, traumatic memory might worsen the overall feeling of hurt and losing control, hence the defensive use of avoidance strategies (Brewin, 2018)”

  1. - A more critical discussion around the gendered implications of sexual abuse would further add to the impact of this paper. For example, discussion of masculine gender norms, taboo and stigmatisation of male sexual abuse, etc would provide this paper with more novelty as men's experiences of sexual abuse as opposed to sexual abuse as a general topic

Response: We thank the reviewer for his/her suggestion. A whole section was added into the discussion to discuss the gendered implications of sexual abuse.

  1. The authors do well in engaging with literature to support the recovery/resilience process of these men's stories. However, it would be great to see more discussion of how the results of this study compare (either in support of or contradiction) to other studies

Response: We thank the reviewer for his/her comment. Unfortunately, most studies we found about resilience regarding male sexual abuse seemed to focus on the breaking of the “victim-offender cycle”. As we were trying to provide new insights, we did not collect data on this specific concept. Therefore, comparing with these findings seems difficult.

  1. Overall, a very interesting paper that sheds light and gives voice to men's lived experiences of sexual abuse (a taboo/stigmatised issue).

We thank the reviewer for his/her nice comment.

Reviewer 2 Report

Overall this makes for an interesting manuscript and will be a useful contribution to the literature. That said, there are a few areas of revision/additions needed in the introduction and methods sections before the article can be accepted. 

Introduction

Whilst succinct the introduction does deal with a number of relevant sources which underpin the study objectives. That said, the authors should seek to provide a more comprehensive review of recent relevant literature currently missing from the introduction. Specifically, (around line 39 onwards) the authors should make reference to recent studies by Agata Debowska and colleagues who have looked at the consequences of abuse on boys separately to girls.

You also need to refer to the work of Siobhan Weare (line 63 onwards) who has extensively interviewed men about their experience of sexual violence and abuse who is not cited in your review of the literature (e.g. Weare 2018, DOI 10.1177/0886260518820815).

You would also benefit from making brief mention of literature and research surrounding sexual violence against women where comparisons to male sexual victimisation have been made. Making reference to recent articles around line 55 would be beneficial to ensure broader links to the sexual violence victimisation literature as a whole are made (refer to Hudspith et al 2023, DOI 10.1177/15248380211050575; Lilley et al 2023, DOI 10.3390/socsci12010034; Sowersby et al 2022, DOI 10.3389/fpsyg.2022.867991).

Methods

You should directly make reference to where ethical approval for the study was gained from and expand on the ethical considerations and procedural steps you undertook given the sensitive nature of the study.

There are issues throughout the manuscript with different use of font size. Be consistent in your use of a single font size throughout

Author Response

We would like to thank the reviewer for his/her help to improve our manuscript. We are very grateful. For ease of review, changes have been in green in the text.

Reviewer 2: Overall this makes for an interesting manuscript and will be a useful contribution to the literature. That said, there are a few areas of revision/additions needed in the introduction and methods sections before the article can be accepted.

Response: We thank the reviewer for his/her nice comment.

  1. Introduction

Whilst succinct the introduction does deal with a number of relevant sources which underpin the study objectives. That said, the authors should seek to provide a more comprehensive review of recent relevant literature currently missing from the introduction. Specifically, (around line 39 onwards) the authors should make reference to recent studies by Agata Debowska and colleagues who have looked at the consequences of abuse on boys separately to girls

Response: We thank the reviewer for his/her suggestion. We added the following sentence: “In a recent study including children and adolescents from Barbados and Grenada, rates of sexual abuse outside and within the family were higher for boys than girls (Debowska et al, 2018). Despite a significant prevalence, it is possible that male sexual abuse remains highly underestimated, as few men report sexual abuse.”

  1. You also need to refer to the work of Siobhan Weare (line 63 onwards) who has extensively interviewed men about their experience of sexual violence and abuse who is not cited in your review of the literature (e.g. Weare 2018, DOI 10.1177/0886260518820815)

Response: We thank the reviewer for his/her suggestion. The reference was added into the section: “In a study focusing on men who were forced to penetrate women, Weare (2018) shows that this form of female-to-male sexual abuse often results in anxiety, depressive and/or suicidal thoughts, self-harm behaviours, mistrust, and negative feelings of anger, shame and isolation”. We also mentioned her work in the last section of our discussion: “According to Weare (2018), “even if some negative impacts are recognised, the gendered construction of masculinity allows for the effects on men to be minimised”. Hence, this research means to report on male sexual abuse impact and tries to propose suggestions to improve their mental health.”

  1. You would also benefit from making brief mention of literature and research surrounding sexual violence against women where comparisons to male sexual victimisation have been made. Making reference to recent articles around line 55 would be beneficial to ensure broader links to the sexual violence victimisation literature as a whole are made (refer to Hudspith et al 2023, DOI 10.1177/15248380211050575; Lilley et al 2023, DOI 10.3390/socsci12010034; Sowersby et al 2022, DOI 10.3389/fpsyg.2022.867991)

We thank the reviewer for his/her suggestion. The references were added into the section: “Hudspith et al (2023) add that rape myths may generate feelings of self-blame and shame which dissuade victims, male and female alike, from revealing the abuse. Amongst common gender-based rape myths, one reads “only gay men are raped; heterosexual men are not” (Lilley and all, 2023). Such false beliefs reinforce the stigma induced by norms of masculinity.” Sowersby et al were cited with reference to this sentence: “Easton et al [7] estimate that it would take an average of twenty years for a man to talk about his history of sexual abuse, due to the stigma induced by current norms of masculinity.”

  1. Methods

You should directly make reference to where ethical approval for the study was gained from and expand on the ethical considerations and procedural steps you undertook given the sensitive nature of the study

We thank the reviewer for his/her remark. Necessary changes were made.

  1. There are issues throughout the manuscript with different use of font size. Be consistent in your use of a single font size throughout

Response: We thank the reviewer for his/her vigilance. There should be no issues with the manuscript now.

Reviewer 3 Report

This scientific article is of extreme importance as very few articles are published on males who have experienced gender based sexual violence. I thank the authors for meeting with the participants, and allowing them to hear their voice. I thank the participants for taking the risk to share their experience. Taking the risk because indeed, trusting another human being is not easy after have been abused.

Finally, I indicated that the conclusion could be improved a little. I believe it would be important to compare and contrast the results from this study with the experience of women, i.e. how is the suffering from men who experienced gender based sexual violence different or similar to that of women? Thank you for addressing this in your conclusion.

It is extremely difficult to recruit male victims of gender-based sexual violence, and in and of itself this is a huge contribution. To hear their voices, what helps them and what they need. I believe this is what the authors tried to do. Yes, they do address the main question, i.e. the broadening of the knowledge and understanding of the consequences associated to gender-based sexual violence. I notice that the authors use the term “male victims of sexual abuse” whereas it might be more appropriate to use gender-based sexual violence. I encourage them to indicate why the have chosen one term over the other.

I find the table most interesting. I notice the aggressors were all known by their victims.  It would have been important to comment more on this and how this might have impacted on the victims’ experiencing of the impact of the abuse and their experiencing of resilience. Same comment re Judicial proceedings.

 I would have liked the authors to discuss how men’s experience, based on the participants’ narratives, compares and contrasts with that of women.

The references are appropriate.

Author Response

We would like to thank the reviewer for his/her help to improve our manuscript. We are very grateful. For ease of review, changes have been in green in the text.

Reviewer 3: This scientific article is of extreme importance as very few articles are published on males who have experienced gender based sexual violence. I thank the authors for meeting with the participants, and allowing them to hear their voice. I thank the participants for taking the risk to share their experience. Taking the risk because indeed, trusting another human being is not easy after have been abused.

Response: We thank the reviewer for his/her nice comment.

  1. Finally, I indicated that the conclusion could be improved a little. I believe it would be important to compare and contrast the results from this study with the experience of women, i.e. how is the suffering from men who experienced gender based sexual violence different or similar to that of women? Thank you for addressing this in your conclusion

Response: We thank the reviewer for his/her suggestion. We addressed the matter into a new section of our discussion regarding the gendered implications of sexual abuse, based on reviewer 1’ suggestion. We added the following sentences about the experience of women: “Studies show that women may suffer from many of the same difficulties as men do: sexual dysfunctions, damaged self-image, feelings of shame and guilt, depression, anxiety, feeling of isolation, relationship-building difficulties… (Pulverman et al, 2018; Browne & Finkelhor, 1986; Banyard et al, 2001)”

The mentioned references were added to the section.

  1. It is extremely difficult to recruit male victims of gender-based sexual violence, and in and of itself this is a huge contribution. To hear their voices, what helps them and what they need. I believe this is what the authors tried to do. Yes, they do address the main question, i.e. the broadening of the knowledge and understanding of the consequences associated to gender-based sexual violence.

Response: We thank the reviewer for his/her comment.

  1. I notice that the authors use the term “male victims of sexual abuse” whereas it might be more appropriate to use gender-based sexual violence. I encourage them to indicate why the have chosen one term over the other.

We thank the reviewer for his/her remark. Since this research aims to highlight the specific issue of male sexual abuse, it seemed important to us to define the participants as male in this study. Hence, we chose to employ “male victims of sexual abuse” instead of “gender-based sexual abuse”.

  1. I find the table most interesting. I notice the aggressors were all known by their victims.  It would have been important to comment more on this and how this might have impacted on the victims’ experiencing of the impact of the abuse and their experiencing of resilience. Same comment re Judicial proceedings

We thank the reviewer for his/her suggestion. We added reflections on the first matter into 2.1 Impact – Relationships with others: “It seems important to note that aggressors were all known by their victims. It is reasonable to assume that experiencing abuse at the hands of trusted peers or family members could induce a profound feeling of betrayal and insecurity. This, in turn, might complicate the building of future, sincere relationships by creating fears, defensive strategies and long-lasting distrust.”

We added reflections on the second matter into 2.1 Impact – Self-fulfilment: “Seven participants engaged in judicial proceedings. At the time of the study, only three aggressors were convicted; two aggressors were not convicted; one case was dismissed; and one case was still ongoing. Some participants may have relied on society to recognize their painful experience and hold the aggressor accountable. Being confronted to different outcomes may reinforce feelings of injustice, anger, stigmatisation and isolation.”

  1. I would have liked the authors to discuss how men’s experience, based on the participants’ narratives, compares and contrasts with that of women.

Response: We thank the reviewer for his/her comment. We tried to underline those questions in the last section of our discussion, based on reviewer 1’ suggestion: “For both women and men, gender roles develop from a very young age. As mentioned before with regards to this specific research, masculine norms would include aggression, rejection of “feminine” characteristics, stoicism, preoccupation with sex, being an economic provider, sexuality and being the protector of home and family (Kia-Keating et al, 2005). Some of the reasons why a male victim of sexual abuse may not talk include the feeling of stigmatisation induced by those norms, as well as the fear of being blamed or rejected (Easton, 2014).

Sexual abuse is harmful regardless of the victim’s gender, and remains a sensitive topic, if not a taboo, amongst society.

Studies show that women may suffer from many of the same difficulties as men do: sexual dysfunctions, damaged self-image, feelings of shame and guilt, depression, anxiety, feeling of isolation, relationship-building difficulties… (Pulverman et al, 2018; Browne & Finkelhor, 1986; Banyard et al, 2001). As the current study aimed to focus on men only, further research should explore the specific impact of gender-based sexual abuse on women’s image of their own femininity and men’s image of their own masculinity.

Nevertheless, with reference to male gender socialisation, current understandings of masculinity leaves little place for understanding of male sexual abuse. It also complicates their own comprehension and acceptance of the abuse and related emotions. According to Weare (2018), “even if some negative impacts are recognised, the gendered construction of masculinity allows for the effects on men to be minimised”. Hence, this research means to report on male sexual abuse impact and tries to propose suggestions to improve their mental health.”

Reviewer 4 Report

First of all, I would like to explicitly thank the authors - this is extremely important work. Research on sexual violence against men or boys is very rare so far and therefore an important contribution.
Nevertheless, I have to say right away that I cannot recommend this paper for publication in the form in which it was written and conceptualised, unfortunately (!). This is primarily due to the sample and the potential of the research questions - 7 of the 10 men interviewed had experienced sexual violence during childhood and/or adolescence, three as adults (from the former partners). This mix is very difficult and actually not acceptable. In the first case, we are dealing with severe child abuse ( where guardians have not fulfilled their duty of care) and in the second case with violence in partnerships. These are far too different contexts to bring them together here. I recommend staying with the 7 participants and focusing more on sexual violence against male children and adolescents. Anything else seems too eratic. I strongly encourage the authors to continue with the data and simply focus the topic more adequately on the experiences of those interviewed.

Author Response

We would like to thank the reviewer for his/her help to improve our manuscript. We are very grateful. For ease of review, changes have been in green in the text.

Reviewer 4: First of all, I would like to explicitly thank the authors - this is extremely important work. Research on sexual violence against men or boys is very rare so far and therefore an important contribution.

Response: We thank the reviewer for his/her comment.

  1. Nevertheless, I have to say right away that I cannot recommend this paper for publication in the form in which it was written and conceptualised, unfortunately (!). This is primarily due to the sample and the potential of the research questions - 7 of the 10 men interviewed had experienced sexual violence during childhood and/or adolescence, three as adults (from the former partners). This mix is very difficult and actually not acceptable. In the first case, we are dealing with severe child abuse ( where guardians have not fulfilled their duty of care) and in the second case with violence in partnerships. These are far too different contexts to bring them together here. I recommend staying with the 7 participants and focusing more on sexual violence against male children and adolescents. Anything else seems too eratic. I strongly encourage the authors to continue with the data and simply focus the topic more adequately on the experiences of those interviewed.

Response: We would like to thank the reviewer for this pertinent comment.

We have removed participants who experienced sexual abuse as adults. We have modified the entire text to maintain consistency with this major change. Verbatims attributed to individuals have been removed. Discussion has also been modified.